# Application of diceCT to Study the Development of the Zika Virus-Infected Mouse Brain

**DOI:** 10.3390/v16081330

**Published:** 2024-08-20

**Authors:** Amy L. Green, Evangeline C. Cowell, Laura M. Carr, Kim Hemsley, Emma Sherratt, Lyndsey E. Collins-Praino, Jillian M. Carr

**Affiliations:** 1College of Medicine and Public Health, Flinders University, P.O. Box 2100, Adelaide, SA 5001, Australia; amygreen58@gmail.com (A.L.G.); evie.cowell@flinders.edu.au (E.C.C.); kim.hemsley@flinders.edu.au (K.H.); 2Flinders Health and Medical Research Institute, Flinders University, Adelaide, SA 5001, Australia; 3School of Biomedicine, The University of Adelaide, Adelaide, SA 5005, Australia; laura.carr@adelaide.edu.au (L.M.C.); lyndsey.collins-praino@adelaide.edu.au (L.E.C.-P.); 4School of Biological Sciences, The University of Adelaide, Adelaide, SA 5005, Australia; emma.sherratt@adelaide.edu.au

**Keywords:** Zika virus, brain imaging, microCT, neurodevelopment, diceCT

## Abstract

Zika virus (ZIKV) impacts the developing brain. Here, a technique was applied to define, in 3D, developmental changes in the brains of ZIKV-infected mice. Postnatal day 1 mice were uninfected or ZIKV-infected, then analysed by iodine staining and micro-CT scanning (diffusible iodine contrast-enhanced micro-CT; diceCT) at 3-, 6-, and 10-days post-infection (dpi). Multiple brain regions were visualised using diceCT: the olfactory bulb, cerebrum, hippocampus, midbrain, interbrain, and cerebellum, along with the lens and retina of the eye. Brain regions were computationally segmented and quantitated, with increased brain volumes and developmental time in uninfected mice. Conversely, in ZIKV-infected mice, no quantitative differences were seen at 3 or 6 dpi when there were no clinical signs, but qualitatively, diverse visual defects were identified at 6–10 dpi. By 10 dpi, ZIKV-infected mice had significantly lower body weight and reduced volume of brain regions compared to 10 dpi-uninfected or 6 dpi ZIKV-infected mice. Nissl and immunofluorescent Iba1 staining on post-diceCT tissue were successful, but RNA extraction was not. Thus, diceCT shows utility for detecting both 3D qualitative and quantitative changes in the developing brain of ZIKV-infected mice, with the benefit, post-diceCT, of retaining the ability to apply traditional histology and immunofluorescent analysis to tissue.

## 1. Introduction

Zika virus (ZIKV) is a positive-sense single-stranded RNA virus in the *Flavivirus* genus, responsible for clinically significant outbreaks that culminated in the 2016 World Health Organisation (WHO) declaration of a Public Health Emergency of International Concern [1,2]. Vertical transmission of ZIKV from mother to child in utero is the primary clinical concern, with adverse consequences for the developing foetus [3,4,5]. The neurological and developmental effects of foetal ZIKV infection are termed congenital Zika syndrome (CZS) [6]. The most common and identifiable pathology in human neonates is microcephaly, defined clinically as a reduction in head circumference [6,7,8], although neurological deficits are also observed in the absence of microcephaly [5,9,10,11,12,13]. Radiological findings, such as by magnetic resonance imaging (MRI) or X-ray computed tomography (CT) scanning of the brain of ZIKV-affected babies, show some common abnormalities, such as calcifications in various regions of the brain [11,14,15,16], ventriculomegaly, and reduced gyrification (i.e., a reduction in the folding of the surface of the brain) [17]. In addition, less common brain pathologies include cerebellar hypoplasia or dysplasia, periventricular septation, and hydrocephalus, highlighting the diversity of CZS [18]. There is also ocular involvement, with findings such as retinal scarring and mottling [6,19,20]. Some of the diversity observed between CZS cases may relate to the timing of ZIKV infection during pregnancy [5,21].

The molecular basis for CZS has been informed by many studies demonstrating ZIKV infection and cell killing of neural progenitor cells in vitro and in vivo [22,23,24,25]. Studies also suggest that, in addition to neurones, astrocytes and microglia are susceptible to infection and viral induction of cell death and contribute to ZIKV-induced brain pathology [22,26,27]. Additionally, mouse models of ZIKV infection during pregnancy demonstrate dramatic effects on the brain, such as reduction in overall size, but often utilise immunodeficient mice, such as those lacking the interferon response, with subsequent severe brain impact and foetal demise [28]. This may reflect aspects of ZIKV infection in utero associated with foetal loss in humans, but, importantly, many babies that are exposed to ZIKV in utero ultimately survive, with varying degrees of impairment at birth and evolving neurological impact [9,15]. Immunocompetent mouse models of direct brain infection of the embryo are established and also demonstrate impact on the brain, measured by both 2D image analysis and histology, such as areas of necrosis and cellular infiltrates in the cortex, hippocampus, thalamus, and cerebellum at 21 days postnatally [25]. Immunocompetent mouse models of early postnatal (P1-P3) ZIKV infection result in acute ZIKV replication and inflammation 3–6 days post-infection (dpi) [29] and demonstrate infection of the retina [30], as well as diverse cell types and regions of the brain. This includes astrocytes and neurones, with infection of the hippocampus and cerebral cortex having the highest infected cell numbers in the cortex [31]. There are subsequent morphological changes in the brain that are consistent with the pathology in humans, including calcifications and ventriculomegaly [22,32,33,34]. These changes have been characterised both histologically and by MRI late in the course of disease in mice (e.g., 21–30 dpi and into adulthood) and can be associated with ongoing inflammation, cell death, and reduced volume of the total brain assessed dorsally by gross morphology [22,35]. Our laboratory has used a similar postnatal ZIKV-infection model in immunocompetent mice [29] and tested the applicability of a relatively new research tool, diffusible iodine-based contrast-enhanced computed tomography (diceCT), to undertake a three-dimensional (3D) in situ analysis of the early (3–10 day) morphology of the ZIKV-infected mouse brain during development and prior to the onset of overt disease. This time period in development reflects late in utero development of the brain and eye in humans and a period of rapid growth of the rodent brain [36,37].

diceCT refers to the process of enhancing the contrast of the soft tissue of a specimen via the application of an iodine-based solution prior to CT scanning [38,39]. The use of micro-CT scanning provides high resolution and the visualisation of bone in small laboratory animals, while iodine staining provides the contrast of soft tissue. The diceCT technique currently has major relevance in general morphological studies, for example, in comparative anatomy and evolutionary studies [40,41] and in the digital preservation of museum specimens [42,43]. Additionally, diceCT has proven useful in recent biomedical and clinical applications in order to study anatomical features [44,45], including brain tissue from laboratory animal and human specimens [46,47], but has not yet been applied to study viral pathology. In contrast, MRI is a widely used clinical imaging technique and useful to apply to living animals, although the machines are less available in the laboratory setting due to the costs and expertise required [48] and provide less resolution of the brain structures in small laboratory animals. While MRI has been combined with improved tissue contrast using Mn ions (MEMRI) and applied to the P1-P11 developing mouse brain with excellent resolution and quantitative analysis [49], this still does not overcome the challenges of using MRI in most laboratory settings. diceCT is thus an alternative technique, with microCT scanning generally more available in core imaging facilities and with outputs that allow for the acquisition of high-contrast soft tissue scans and 3D visualisation and analysis of these anatomical structures in situ within the natural restriction of bone. Further, post-diceCT-stained tissue has the potential to be utilised for further downstream analysis, such as general histology, which benefits the research setting.

Here, the application of diceCT to the early postnatal mouse successfully discriminated regions of the brain, the lens, and retinal layers in the eye and could quantitatively measure changes in normal brain development. Further, diceCT could qualitatively identify putative ZIKV-infected animals in an unbiased cohort before the disease is overt (6 dpi) and can both qualitatively and quantitatively identify changes in regions of the brain in ZIKV-infected animals later during infection (10 dpi), when clinical signs are emerging. The changes identified are consistent with previous descriptive pathology in mice and humans, which we further extend with quantitative analysis, showing a decline in tissue volume in multiple regions of the brain. Post-diceCT, histological Nissl, and immunostaining for the microglial marker, Iba1, were successful. Thus, in the future, diceCT may serve as a relatively simple addition to the toolkit for research studies seeking to link in situ tissue morphology to detailed cellular and molecular changes, further facilitating our understanding not only of ZIKV-induced brain pathology but also, potentially, of other neurodevelopmental or neurodegenerative disorders.

## 2. Materials and Methods

### 2.1. ZIKV Stocks and Infection of Neonatal Mice

Viral infections utilised the ZIKV strain PRVABC59, amplified in C6/36 cells with cell culture supernatants harvested, clarified, filtered, and stored at −80 °C. The infectious titre was determined by plaque assay on Vero cells and quantitated as a plaque-forming unit (pfu) per mL, as previously described [29,50].

One-day-old Balb/c pups (*n* = 23 total) were infected by injection into the body above the milk spot with a 31G insulin needle and 5000 pfu of ZIKV, in a total volume of 10 µL, as previously described [29]. For a 25% decrease with a 10% error, α = 0.05, and 80% power, *n* = 3 animals per group is calculated for appropriate statistical power. Mice were monitored daily for colour and movement. At designated time points post-infection (pi), mice were weighed and humanely killed by decapitation. A tissue sample from the neck at the time of decapitation was taken for RNA extraction and RT-PCR for ZIKV, as previously described [29]. Whole heads were fixed in 10% (*v*/*v*) buffered formalin overnight and thereafter stored in phosphate-buffered saline (PBS) at room temperature. Samples were collected at 3- and 6-day pi (dpi) (*n* = 4 uninfected, *n* = 2 ZIKV per time point), across 2 different litters, and at 10 dpi for ZIKV and uninfected animals from the same litter (*n* = 3/group). For unbiased blind analysis at 6 dpi, samples reflected *n* = 3 uninfected and *n* = 4 ZIKV infected within one litter. This study was undertaken in accordance with the Australian Code for the Care and Use of Animals for Scientific Purposes (2013) and with approval from the Flinders University Animal Welfare Committee, AEC BIOMED 4707-6, and the Flinders University Institutional Biosafety Committee, Microbiological approval, 2016-07.e1. Prior use of this model demonstrated no impact of the injection, and hence control mice were left uninfected rather than mock-infected with PBS to minimise handling of the neonates.

### 2.2. DiceCT Staining and Scanning

Fixed specimens were submerged in 3.75% (*w*/*v*) B-lugol’s solution, as described in Dawood et al. (2021), to reduce soft-tissue shrinkage [51]. Even penetration of the iodine stain was facilitated by incubation at room temperature on a rocker for either approximately 36 h (h) for specimens at 3 and 6 dpi or 48 h for 10 dpi specimens. Specimens were rinsed with PBS and prepared for scanning by wrapping them in foam and PBS-soaked gauze to avoid desiccation. Specimens were subjected to X-ray micro-computed tomography (micro-CT) using a Bruker SkyScan 1276 high-resolution micro-CT at Adelaide University Microscopy Facility (Adelaide Health and Medical Centre [AHMS], University of Adelaide, Adelaide, SA, Australia). Settings were as follows: 90 kV, 200 μA, 430 milliseconds (ms) exposure time, frame averaging of 2, 0.2° rotation steps, and using a 0.25 mm aluminium filter. Each specimen was scanned for approximately 90 minutes (min) for a final resolution of 6.5 μm (detector resolution at 4032 × 2600 pixels). The radiographs from these scans were reconstructed in NRecon v.2.0 (Bruker, Kontich, Belgium) to acquire 2D cross-section images, or ‘slices’. The slices were processed in DataViewer, 3D inspection software (Bruker, Kontich, Belgium) to allow for the alignment of specimens’ X, Y, and Z axes.

### 2.3. Image Segmentation, Visualisation, and Quantitation of Brain Volumes

Brain segments were defined based on labelled regions marked in the Allen Developing Mouse Brain Reference Atlas [52]. To acquire volume measurements of the whole brain and brain segments, regions were digitally segmented from CT scans with 3D slicer (www.slicer.org, accessed March 2023) [53] to create 3D models. The add-on module SlicerMorph [54] was used to import the aligned CT scans into the 3D slicer programme using the ‘ImageStacks’ function. This allowed the CT scans to be loaded into the programme at a lower resolution to allow efficient segmentation on a general-use computer system. This was appropriate for the easily identifiable brain regions analysed here but would not be recommended when segmenting less visible or finer structures, such as the retina. Three-dimensional Slicer’s ‘Reformat’ module was also used to visually reorient images where necessary. The ‘Segment Editor’ module was used to digitally segment regions of the brain. A region was defined using the ‘+’ function to name and select region colour, and the ‘Paint’ tool was used to manually select the region of interest. Regions were manually selected in slices across the X, Y, and Z axes, and the ‘Fill between slices’ tool was used to interpolate defined regions between slices. The interpolated region between the manually defined slices was dependent on the rate of change in the region across the number of slices and varied for each region. A new manual definition of a region was undertaken when it was determined that the visual CT region no longer fit the previously manually defined region. All manual segmentation was undertaken by one operator to ensure consistency in segmentation. To obtain the 3D models, the defined segments within the ‘Data’ module were selected, and ‘export visible segments to models’ was chosen. The generated segmentation models could then be exported from a 3D slicer for analysis. Models exported from 3D slicer were edited in MeshLab v.2021.07 [55] using the ‘Surface Reconstruction: Screened Poisson’ function [56] to ensure they were manifold and that the volumes of the whole brain and separate regions could be correctly quantified. Volume was calculated using the package Rvcg v.0.22.1 function ‘vcgVolume’ in the R Statistical Environment v.4.3.1 [57]. Slicer data sets are available on Figshare (https://figshare.com/articles/dataset/diceCT_imaging_of_the_ZIKV_mouse_brain_during_development/26086651, edited 19 August 2024).

### 2.4. Blinded Review of 6 dpi Scans

A link to rendered videos from coronal sections was numbered 1–7 and provided to four authors for review: two virologists (ECC, JMC) and two rodent neuroanatomists (LMC, LC-P). The author (ALG), who undertook the diceCT analysis, was excluded from the review. Videos were viewed independently and assigned as uninfected, ZIKV-infected, or undetermined, with supporting comments provided.

### 2.5. Tissue Destaining and Preparation of Sections

Following diceCT scanning, tissues were destained by submersion in 1% (*w*/*v*) sodium thiosulphate on an orbital platform for 3 days to remove the iodine stain, followed by 2 days in deionised water to remove the destaining reagent [58]. Destaining was essential for obtaining good-quality tissue sections. Destained tissue was serially treated with 10%, then 20% (*w*/*v*) sucrose in PBS for 1 h each, and finally transferred to 30% (*w*/*v*) sucrose in PBS overnight or until the tissues sank to the bottom of the tube. Once cryoprotected, tissues were embedded (OCT compound, TissueTek). 15 μm coronal cryostat sections were mounted onto glass slides and air dried for 30 min at room temperature, then stored in an airtight container at −20 °C.

### 2.6. Immunolabelling and Nissl Staining of Post-diceCT Tissue

Frozen embedded tissue sections were thawed and then rehydrated in PBS for 10 min at room temperature. Antigen retrieval was performed for 20 min in 10 mM sodium citrate, 0.05% Tween 20, pH 6, that was heated until boiling. Tissue sections were permeabilised with 0.5% (*v*/*v*) IGEPAL (Sigma-Aldrich, St. Louis, MO, USA) in PBS for 20 min at room temperature, then rinsed 3 times for 5 min in PBS. Non-specific binding was blocked with 5% (*v*/*v*) normal goat sera (NGS) and 2% (*w*/*v*) bovine serum albumin (BSA) in PBS for 1 h at room temperature. Tissue sections were immunostained with anti-ionised calcium binding adaptor molecule 1 (Iba1) (rabbit polyclonal, 1:100 dilution, 1 μg/mL, Wako) diluted in 2% (*v*/*v*) NGS in PBS in a humidified chamber for 24 h at 4 °C. Sections were washed in PBS and bound antibody detected using anti-rabbit Alexa Fluor 555 (ThermoFisher Scientific, Waltham, MA USA), diluted in 2% (*v*/*v*) NGS in PBS for 1 h at room temperature, protected from the light. Autofluorescence was quenched (ReadyProbes™ tissue autofluorescence quenching kit, Thermofisher) by post-staining incubation with quenching reagent for 5 min at room temperature. Nuclei were then stained (Hoechst 33342, 5 μg/mL, ThermoFisher Scientific) for 10 min at room temperature and protected from the light. Sections were then mounted (ProLong Gold Antifade, ThermoFisher Scientific) and imaged by fluorescent microscopy (VS200 Slide Scanner, Olympus, Tokyo, Japan).

Additional serial sections were selected for cresyl violet staining of the Nissl body. Fixed sections were air dried for 1 h at room temperature, then de-fatted by immersion in a 1:1 chloroform–ethanol solution for 3 h. Sections were rehydrated through a series of ethanol dilutions: 100%, 95%, and 75% (*v*/*v*), and then deionised water for 3 min. The sections were stained for 10 min at 37 °C in 0.1% (*w*/*v*) cresyl violet, then rinsed under gently running deionised water. Staining was differentiated using 95% ethanol, then dehydrated in 100% ethanol, 2 × 5 min, and cleared in xylene, 2 × 5 min. Stained sections were mounted in DePeX mounting medium and examined using brightfield microscopy (VS200 Slide Scanner, Olympus).

### 2.7. Statistical Analyses

Measurements of mouse body weight were normally distributed (Shapiro–Wilk test) and compared using Student’s unpaired *t*-test. Volume data of the whole brain and separate regions were analysed for statistical differences between treatment groups and days post-infection in the R Statistical Environment using a one-way analysis of variance (ANOVA) with Tukey’s multiple comparison test. Significance was set at an alpha level = 0.05. Results were visualised using GraphPad Prism v. 10.0.0.

## 3. Results

### 3.1. DiceCT Can Track Brain Development and the Impact of ZIKV Infection

Newborn P1 mice were left uninfected or ZIKV-infected. Animals were taken at 3 and 6 dpi, where we know there is increasing viral replication and induction of antiviral and inflammatory host responses with a lack of overt clinical signs [29]. This infection model was also extended to 10 dpi with a single litter to avoid across-litter variation, where half of the litter was left uninfected and half ZIKV-infected, and analysis was performed in a blind fashion. Mice that were determined retrospectively to be ZIKV positive by RT-PCR were visibly smaller, with a significantly lower total body weight (*p* = 0.007, t = 5.042, df = 4) and observable mild hind limb dysfunction, presenting as a mild tremor or reduced ability to traverse a straight line (Table 1). RT-PCR was undertaken as a means to identify ZIKV-positive animals in a non-quantitative manner on tissue from the neck at the time of the humane killing in order to preserve the brain intact for diceCT analysis. Head measurements tended to be lower but were not significantly reduced (Table 1). DiceCT staining was performed on whole heads, which improved the resolution of regions of the brain, particularly the frontal and mid-region, compared to microCT scanning alone (Figure 1A,B). DiceCT analysis yielded clear discrimination of regions of the developing mouse brain across P4–P11 in uninfected and infected animals (3–10 dpi) (Figure 1B). At this time point, the eyes are still closed, and the lens and retinal layers are clearly visible on imaging. The developing teeth are evident. In general, no clear defects were observed in the ZIKV-infected mouse brain at 3 dpi based on visual inspection, although a unilateral gap in the cortical region was visualised in one animal, as shown (Figure 1B). By 6 and 10 dpi, unilateral and bilateral gaps, often at the point of intersection of different regions of the brain and most apparent at the level of the posterior hippocampus, as well as expansion of ventricular spaces, were more widely observed in ZIKV-infected animals. Serial coronal images are shown across the brain of one animal at 10 dpi, with clear pathology in the olfactory bulb and frontal regions of the brain (Figure 1C). Overall, the ZIKV-induced brain pathology was diverse, and no hallmark or characteristic pathology was consistently identified across all ZIKV-infected animals. There was also no evidence of calcifications in ZIKV-infected mice identified by gross morphology at this early time of infection.

Image outputs were subjected to quantitative analysis. Software programmes CTAn and CTvol (Bruker-microCT, Kontich, Belgium), Drishti (Limaye, 2012), Avizo (Thermofisher Scientific), VGStudio Max (Volume Graphics, Heidelberg, Germany), Imaris (Oxford Instruments, Abingdon, UK), and 3D slicer were assessed for ease of use and availability for a routine virology research laboratory in quantitative analysis of CT images. 3D slicer was chosen as an intuitive, free, open-source programme that can handle large data sets without specialised computing power and has good online supporting tutorial resources [53]. The micro-CT files were converted to produce high-definition 3D outputs that allow visualisation and quantitation of the surface volumes and internal tissue detail to undertake manual and semi-automated segmenting. Scans were reconstructed, and the resulting images were segmented and analysed [53], with confident segmentation of the brain into seven regions: the olfactory bulb, cerebrum, hippocampus, interbrain (comprising thalamus and hypothalamus), midbrain, cerebellum, and hindbrain (including brain stem). A representative schema illustrating the segmented regions is shown (Figure 2A,B). Quantitation of whole brain volume demonstrated a significant increase over developmental time in uninfected mice, as expected (*p* = 0.0001, F = 68.5, df (5, 5)). There was a significant increase from 3 to 6 dpi in ZIKV-infected mice (*p* = 0.0008) and a significant decline between 6 and 10 dpi (*p* = 0.0006) in total brain volume (Figure 2C). Total brain volume was not significantly different at 3 and 6 dpi for ZIKV-infected compared to uninfected mice, but total brain volume was reduced at 10 dpi in ZIKV-infected mice relative to uninfected mice at the same time point (*p* = 0.0002) (Figure 2C).

The same pattern of growth and impact of ZIKV was seen in other regions of the brain (Figure 3). In the cerebrum (*p* < 0.0001, F = 67.12, df (5, 10)) and olfactory bulb (*p* < 0.0001, F = 36.10, df (5, 10)), volumes increased from 3, 6 to 10 dpi in uninfected mice. Volumes similarly increased from 3 to 6 dpi in ZIKV-infected mice, but there was a significant decline in volume by 10 dpi (*p* < 0.0001; *p* = 0.0181, respectively). Cerebrum and olfactory bulb volume were also significantly reduced in ZIKV-infected compared to uninfected mice at 10 dpi (*p* < 0.0001; *p* = 0.0011, respectively) (Figure 3). In the hippocampus (*p* = 0.0030, F = 7.916, df (5, 10)), interbrain (*p* = 0.0013, F = 9.902, df (5, 10)), and cerebellum (*p* < 0.0001, F = 45.68, df (5, 10)), volumes also increased from 3 to 10 dpi in uninfected animals and were reduced in ZIKV-infected mice at 10 dpi compared to uninfected mice at the same time point (*p* = 0.0383; *p* = 0.0100; *p* = 0.0004, respectively) (Figure 3). For the midbrain (*p* = 0.0024, F = 8.384, df (5, 10)) and hindbrain (*p* = 0.0514, F = 3.291, df (5, 10)), there was no significant increase in volume over time in uninfected animals. For the midbrain, uninfected mice at 10 dpi show no increase in volume compared to 6 dpi and a significant reduction in ZIKV-infected mice at 10 dpi compared to uninfected mice at this same time point (*p* = 0.0098). For the hindbrain, no volume changes were significant, although notably, one ZIKV-infected sample at 6 dpi was noted post-diceCT analysis to be missing the brain stem, presumably during the collection process (Figure 3).

Although quantitative brain measurements at 6 dpi were not significantly different between uninfected and ZIKV-infected animals, our qualitative analysis suggested that defects might be visually evident at this time point, although with diverse pathologies. Given this, the ability to utilise diceCT to visually or quantitatively identify ZIKV-infected animals in a blind cohort at 6 dpi before the disease is clinically overt was assessed. Neonatal P1 mice from a single litter were left uninfected (*n* = 3) or ZIKV-infected (*n* = 4) and, at 6 dpi, were subjected to diceCT analysis. Independent visual analysis of images by four reviewers with diverse expertise in virology and rodent neuroanatomy identified consistent morphological differences in animal numbers 2, 4, and 6 that penetrated multiple layers of the brain (Figure 4, Table 2). All reviewers identified animal number 1 definitively as uninfected. Three of the four authors identified animal 5 as ZIKV-infected but with some reservations, and similarly, animals 3 and 7 were deemed to be likely uninfected or presented some uncertainty (Table 2). Quantitative analysis was performed on segmented brain images, per Figure 2. In line with the blind ratings, animal 4 tended to reflect smaller measures in the interbrain, midbrain, and cerebellum. Brain volumes, however, were not significantly different across the cohort, and samples did not cluster well into two distinct groups that could suggest alignment with either uninfected or ZIKV-infected animals (Table 3). The biggest variability was observed at the level of the posterior hippocampus, interbrain, and midbrain, both visually and in the quantitative analysis. The identification of animals as uninfected or ZIKV-infected could not be resolved without sampling the brain tissue since ZIKV was undetectable by PCR from the body at this time point pi. Thus, we believe diceCT imaging could qualitatively identify most ZIKV-infected animals with some certainty but cannot quantitatively predict ZIKV infection in a blind cohort prior to the clinical presentation of the disease.

### 3.2. Tissue Can Be Utilised Post diceCT for Traditional Histology and Immunostaining

DiceCT defines tissue structures in 3D spatial relationships, and it would be informative to link this to changes at a cellular level. The utility of the diceCT-processed tissue for subsequent RNA analysis and traditional histological and immunostaining was assessed here. Samples post-diceCT that had been stored in PBS were destained to remove the iodine. Sections of brain cortical tissue were sampled, and RNA was extracted using the traditional Trizol method or a commercial kit designed for RNA recovery from formalin-fixed tissue specimens. Spectrometry analysis of recovered RNA demonstrated poor yields and quality, as judged by A260/280 nm. A trial RT-PCR for the housekeeping gene, GAPDH, on these samples was not successful. Destained tissue was also processed for traditional embedding. Specimens that were not extensively destained prior to embedding ‘shattered’ and intact good-quality sections were not obtained. This was observed for both OCT and paraffin wax-embedded sections. Conversely, however, extensive destaining prior to sucrose treatment and OCT embedding resulted in successful sectioning. Nissl and immunofluorescent staining for the macrophage/microglial marker, Iba1, were both successful (Figure 5A,B). Staining for Iba1 was chosen as an exemplar antigen, as it is widely expressed across multiple regions of the brain, with positive cells having distinct morphology, including round immature microglia and more mature cells with distinct branches and processes [59]. Immunostaining was of appropriate quality to yield good-resolution images but did lack some clarity in some areas (Figure 5B). The soma and processes of Iba1-positive cells in the cortex and hippocampus reflected the expected diversity of morphological microglial classifications at this developmental time point, including both large round cell cells and smaller cells with visually distinct processes. Particularly intense staining of Iba1-positive cells was seen along the edges of the cortex, potentially representing cells in vessels or meninges.

## 4. Discussion

While a lot is known about ZIKV at a cellular level, the relationship between interacting cells and within specific regions of the brain throughout infection and brain development is less well described. The rarity of clinical material from ZIKV-affected brains makes in vivo laboratory models of significant value. The diceCT technique applied here to the developing mouse brain and in the context of ZIKV has provided resolution of different soft tissue regions of the brain that allowed segmentation of these regions with a confidence that could not be achieved by microCT analysis alone. Further, diceCT imaging resolved structures in spatial relationship to each other in situ. Here, diceCT has been applied to analyse the impact of ZIKV infection over early postnatal development (P4, P7, and P11), equating to early infection time points (3, 6, and 10 dpi). Quantitatively, in uninfected mice, there is a clear and significant increase in the volume of the total brain, cerebrum, hippocampus, and olfactory bulb with time, indicating growth of these regions during the P4–P11 stages of development and consistent with a major postnatal increase in brain volume in rodents out to P20 [37]. For the cerebellum, the significant volume increase was between P7 and P11, suggesting a more significant growth phase at this later time. This is consistent with MEMRI quantitation of these regions over the same time period, with a greater rate of volume increase in the cerebellum from P6 [49]. This also aligns with the later development of granule cell (GC) neurones in the hippocampus and olfactory bulb and in the cerebellum in the first 2 weeks postnatally [37,60]. For the midbrain and hindbrain, trends for volume increases are evident, but these did not reach significance and would benefit from increased animal numbers for this quantitative analysis.

At 3 and 6 dpi in ZIKV-infected mice, the virus is replicating, and induction of inflammatory responses is ongoing [29]; however, the mice at this time are physically indistinguishable in behaviour or size and not quantitatively different by diceCT imaging from uninfected mice. Notably, however, morphological defects could be visually detected by assessors with expertise in rodent neuroanatomy in some individual ZIKV-infected animals at 6 dpi. When a litter of uninfected and ZIKV-infected animals was blindly analysed at 6 dpi, three out of four mice were putatively identified by four independent researchers, and a remaining fourth mouse was identified by three out of four researchers as ZIKV-infected. The impact of ZIKV infection became more pronounced with time; at 10 dpi, ZIKV-infected mice were significantly smaller with the onset of clinical signs. Visually, the ZIKV-induced pathology at 6 and 10 dpi was diverse, with changes observed in multiple regions of the brain, as is also reported in ZIKV-infected babies [5,11,12,14]. The ZIKV morphologies observed in the neonatal mouse model included ventriculomegaly, as previously described [30,31,34]. Interestingly, expansion of ventricular spaces was seen at 6 dpi without quantitative reduction in any tissue volume, suggesting this could reflect fluid accumulation and a cellular infiltrate due to an inflammatory response. At 10 dpi, quantitative volume measures in the total brain and all other brain regions, except the hindbrain, were significantly reduced in relation to uninfected mice. The brain volumes in 10 dpi ZIKV-mice were also reduced relative to 6 dpi and were not significantly different compared to 3 dpi ZIKV-infected mice. This is suggestive of tissue loss and is consistent with cell killing, as previously observed for neurones and NPC [25] and with the detection of cleaved caspase 3 and fluoro-Jade in the cortex, hippocampus, and cerebellum of ZIKV-infected neonatal mice [34], as well as with neuronal losses observed previously in ZIKV-infected cells and brains [22,23,25]. The volumes of the cerebrum and cerebellum were reduced in ZIKV-infected animals. Functional loss within the cerebellum or in connections with the cerebral cortex is consistent with the onset of motor dysfunction at 10 dpi in ZIKV-infected mice and aligns with the timing of the development of GC neurones and interneurones in these brain regions [60]. Here, diceCT imaging has not resolved neuronal linkages between regions of the brain, but this could be specifically investigated in the future by immunostaining for neuronal markers or retrograde tracer studies, pending the feasibility of these methods post-diceCT. Quantitative analysis of the midbrain, an important site of progenitor cells, including radial glial cells for neurones in the cortex, was also of significant interest [37]. The analysis demonstrated a trend towards volume increases in the midbrain from 3 to 6 dpi in both uninfected and ZIKV-infected animals. There was no further increase in volume in uninfected mice at 10 dpi, suggesting a reduction in progenitor expansion at this point in development, but midbrain volume decreased significantly in 10 dpi ZIKV-infected animals, again suggestive of cell death. Cell death and neurodegeneration could be specifically assessed, as previously conducted by caspase-3 or fluoro-Jade staining [34], pending compatibility with post-diceCT tissue. The timing of the onset of visual pathology at 6 dpi, but the quantitative loss of brain volumes in multiple regions later at 10 dpi, is mechanistically interesting. This could reflect the cell killing of progenitor cells at 6 dpi, which manifests as reduced brain volumes at 10 dpi due to a lack of progenitor cell numbers to migrate and expand to these multiple brain regions during later development, rather than the viral killing of pre-formed brain regions. Additionally, our qualitative observations were more pronounced in the frontal–midbrain or around the hippocampus, perhaps also reflecting a role of the normal posterior–anterior process of myelination that could protect the developing brain from injury [37]. Changes in the hindbrain and cerebellum are also of future interest since seizures are reported in later stages of this mouse model of ZIKV infection [35], and caudal or rhombencephalon morphologies are strongly associated with epilepsy in CZS [61]. Overall, the findings of the ZIKV-induced reduction in the volume of multiple regions of the brain are consistent with the prior description of diffuse and diverse ZIKV infection in NPC, astrocytes, and neurones across multiple regions of the brain, in vitro and in this mouse model [22,31]. Importantly, while the study here has quantitated changes in some regions of the ZIKV-infected brain at 10 dpi, a greater-powered study with larger numbers of animals may be needed to identify smaller quantitative changes at 3 or 6 dpi or in other regions of the brain.

While our previous work has demonstrated measurable changes in the layers of the retina at 6 dpi in this model by traditional histology [29], no major visually apparent changes in the retina or eye were detected. This remains to be quantitatively assessed. Interestingly, ZIKV has been recently reported to infect and kill cells of the developing ear [62]. The auditory canals can be visualised in our diceCT images, and, hence, this is also an area of interest in the future for a more detailed analysis of this image set to potentially define a laboratory model of hearing loss in ZIKV-affected newborns.

Prior to undertaking diceCT, it was hoped that the iodine contrast could resolve potential calcifications as reported in ZIKV-infected babies, but these findings were not evident in our study, either macroscopically or via diceCT. This is likely due to the formation of calcifications as a result of the longer-term impact of cell death in the brain and is seen at 3–4 months post-infection in a neonatal mouse model of ZIKV infection, microscopically and macroscopically on the surface of the brain [34]. It may be that the timepoints investigated in the current study were too early to see these changes. Alternatively, this inability to detect calcifications may be due to the limitations of the diceCT technique itself, with a loss of contrast for low-density deposits, such as calcium, compared to soft tissue in diceCT tissue. Micro-CT alone may be better for specifically analysing this pathology.

Of note, our study successfully utilised post-diceCT tissues for both traditional histology and immunostaining, including Nissl staining and Iba1 immunostaining. Importantly, however, the tissue must be thoroughly cleared of iodine and destain; otherwise, the sample is brittle and hard to section, leading to the shattering observed in the current work. The exemplar immunostain used in the current work, Iba1, indicated that the tissue may have lost some integrity during the diceCT process, but nonetheless was still able to detect a diversity of microglial morphology, as is expected with embryonic and early postnatal immature proliferative microglia transitioning towards the ramified mature adult state [63,64,65]. Staining was particularly strong in the meninges and could reflect perivascular macrophages or trafficking microglia [66]. While some optimisation of the preparation process may be needed to maximise tissue and antigen integrity, the methodology still offers the benefit of linking 3D changes defined by diceCT to 2D cellular detail without the need for 3D stitching of stained images and with analysis undertaken on the exact same tissue. Similarly, other brain markers, such as NeuN (neurones), GFAP (astrocytes), stem cell markers (e.g., Sox2), or BrDU/Edu proliferation studies, should be tested and will be informative. Unfortunately, however, good-quality RNA was not able to be extracted using the current methodology, limiting the ability to link 3D changes with alterations in molecular gene expression, another area of significant interest, by PCR without further methodological refinement. Techniques such as nanostring analysis, as we have previously performed to assess inflammation in this model [29], might still be appropriate to apply to RNA extracted from post-diceCT tissue.

Importantly, beyond ZIKV, diceCT has broader applicability to study the brain in other mouse neurological models, particularly neurodevelopmental disorders, where imaging in 3D and progressively with time is of interest [67]. Iodine staining and micro-CT analysis are amenable to broad applicability in laboratory settings, with inexpensive and easy staining protocols, as well as the readily available and relative accessibility of micro-CT equipment in many core imaging facilities. One caveat, however, is the inability to undertake longitudinal imaging in the same animal due to the terminal nature of the staining method. Further, the programmes used for quantitation are developing and becoming easier for non-experts to use in a time-efficient manner. The most time-consuming aspect of this study was the segmentation process. To address this, there is the potential to develop automated segmentation using 3D slicer [68]. Artificial intelligence (AI) training of data in relation to a specific model, such as differences between the typically developing mouse and that infected with ZIKV, will empower finer detail and efficiencies in brain segmentation.

Overall, this study has applied diceCT to define early morphological changes in the developing mouse brain and the impact of ZIKV on this. The validity of diceCT in this setting is supported by the consistency of findings with the existing literature, while the added benefit of 3D information and flexibility to align with other laboratory methodologies will assist in the logistics of including diceCT analysis in a modern research plan. Incorporating diceCT will impact our ability to link the overall anatomical pathology to specific cellular and molecular changes in a context that retains the spatial interactions within the 3D of the brain to improve our holistic definition of ZIKV models of pathology and disease.

## Figures and Tables

**Figure 1 viruses-16-01330-f001:**
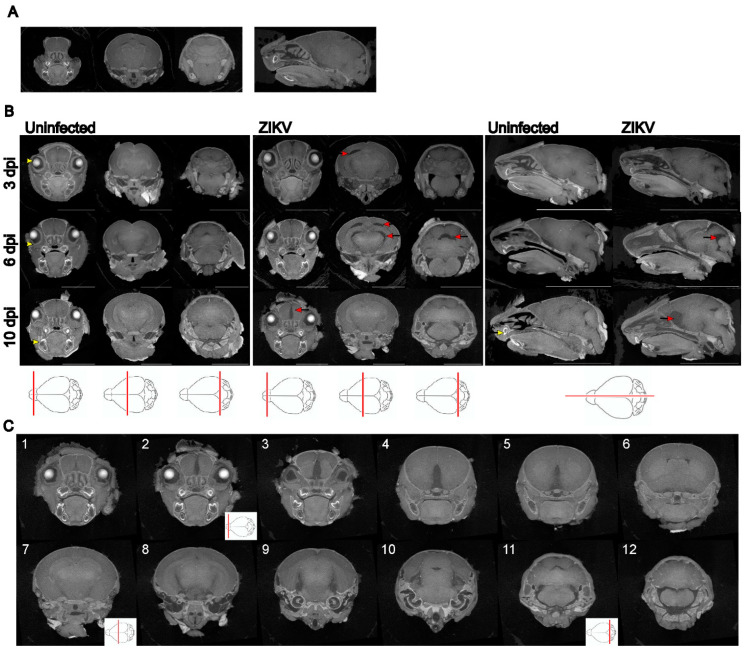
**diceCT imaging of uninfected and ZIKV-infected neonatal mouse heads throughout development.** P1 mice were left uninfected or ZIKV-infected, and at the time points indicated, whole heads were collected and subjected to diceCT. Images across three coronal and one sagittal plane of selected animals are shown (position indicated by red line on brain outline image below), (**A**) microCT on a P7 uninfected animal; *NOTE:* the eyes have been enucleated in these animals; (**B**) from animals at 3 and 6 dpi (*n* = 4 mock, *n* = 2 ZIKV per time point), and 10 dpi (*n* = 3/group). Examples of lenses, retinas, and teeth are indicated by yellow arrows. Red arrows highlight examples of ZIKV-induced pathology. Scale bars: 0.5 cm (coronal); 1 cm (sagittal); 1 (**C**) 1-12 represent serial coronal sections from one 10 dpi ZIKV-infected animal. A video compilation of images is available at Figshare (https://figshare.com/articles/dataset/diceCT_imaging_of_the_ZIKV_mouse_brain_during_development/26086651, edited 19 August 2024).

**Figure 2 viruses-16-01330-f002:**
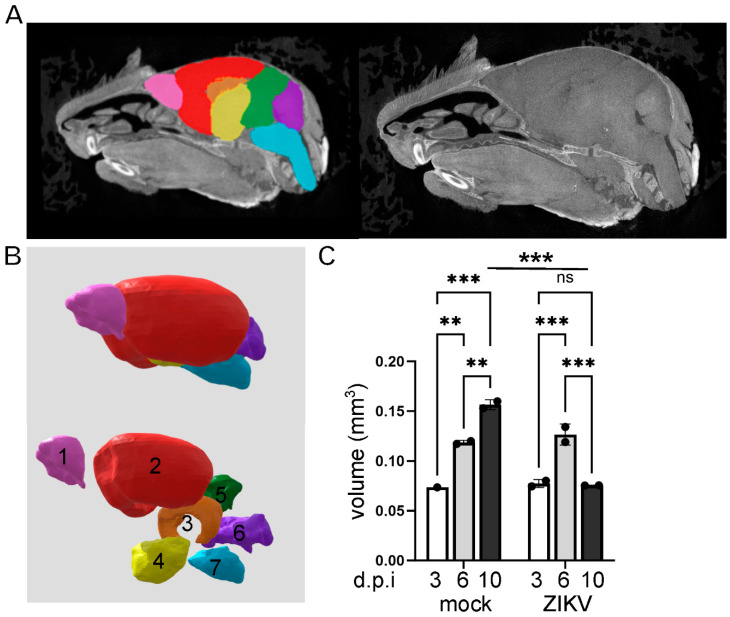
Brain segmentation and quantitation of total brain volumes throughout development in uninfected and ZIKV-infected mice. P1 mice were left uninfected or ZIKV-infected, and, at the time points indicated, whole heads were collected and subjected to diceCT. Images were rendered, brains segmented, and volume quantitated. (**A**) Sagittal diceCT image with segmented regions of the brain superimposed; (**B**) segmented brain regions 1 = olfactory bulb, 2 = cerebrum, 3 = hippocampus, 4 = interbrain (thalamus and hypothalamus), 5 = midbrain, 6 = cerebellum, 7 = hindbrain and brain stem; (**C**) Total brain volume. ** *p* < 0.01; *** *p* < 0.001, one-way ANOVA, Tukey’s multiple comparison test, ns = not significant. 3 and 6 dpi (*n* = 4 mock; *n* = 2 ZIKV per time point), and 10 dpi (*n* = 3/group).

**Figure 3 viruses-16-01330-f003:**
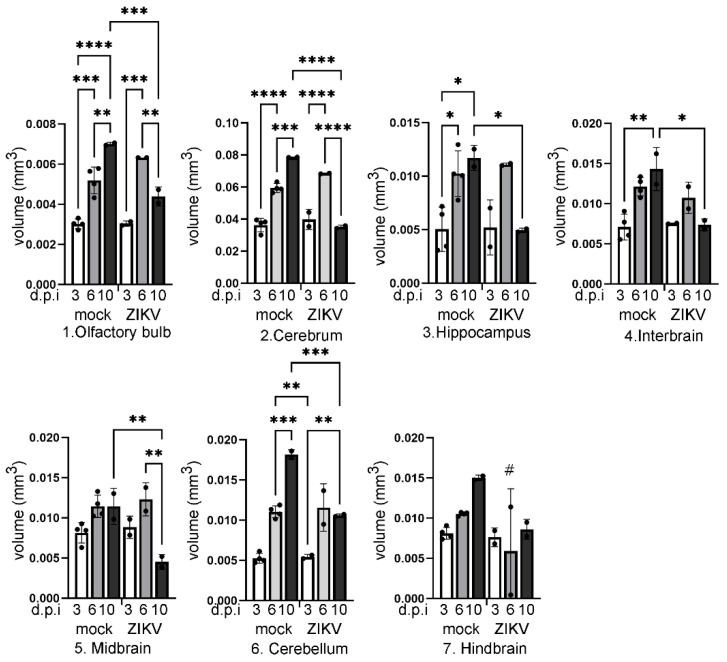
Quantitation of changes in volumes of different regions of the brain throughout development in uninfected (UI) and ZIKV-infected mice. P1 mice were left uninfected or ZIKV-infected, and at the time points indicated, whole heads were collected and subjected to diceCT. Images were rendered, and brain regions 1 (olfactory bulb), 2 (cerebrum), 3 (hippocampus), 4 (interbrain), 5 (midbrain), 6 (cerebellum), and 7 (hindbrain) were segmented as outlined in Figure 2, and volume quantitated. * *p* < 0.05, ** *p* < 0.01, *** *p* < 0.001, **** *p* < 0.0001, one-way ANOVA, Tukey’s multiple comparison test. # = unreliable due to loss of brain stem at sample collection.

**Figure 4 viruses-16-01330-f004:**
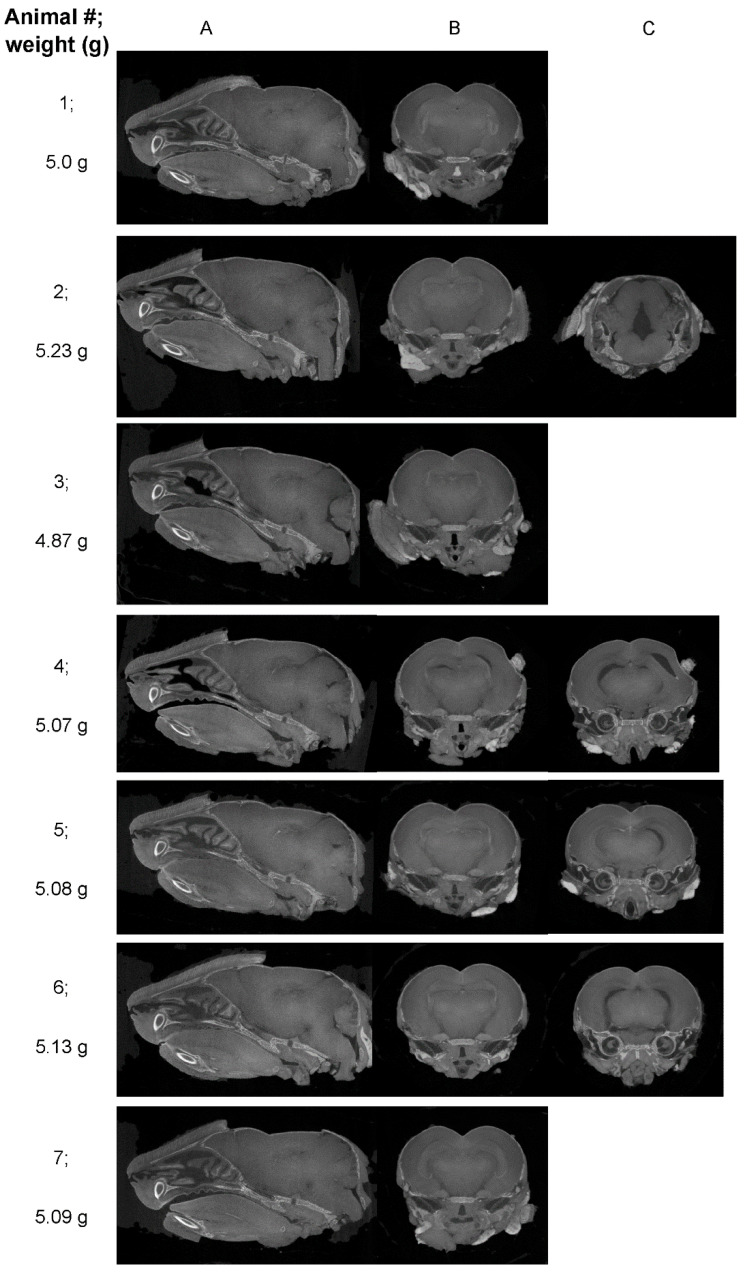
**diceCT imaging of the brain at 6 dpi in a blind cohort of uninfected and ZIKV-infected animals.** P1 mice from a single litter were left uninfected (*n* = 3) or ZIKV-infected (*n* = 4), then at 6 dpi, whole heads were taken and subjected to diceCT in a blinded fashion. (**A**) single sagittal; (**B**) coronal section from each animal is shown, as in Figure 1; (**C**) example regions of identified significant pathology. Numbers (#) 1–7 represent individual mice, and weights (g) at 6 dpi are indicated. Images compiled into videos are available at Figshare https://figshare.com/articles/dataset/diceCT_imaging_of_the_ZIKV_mouse_brain_during_development/26086651, edited 19 August 2024.

**Figure 5 viruses-16-01330-f005:**
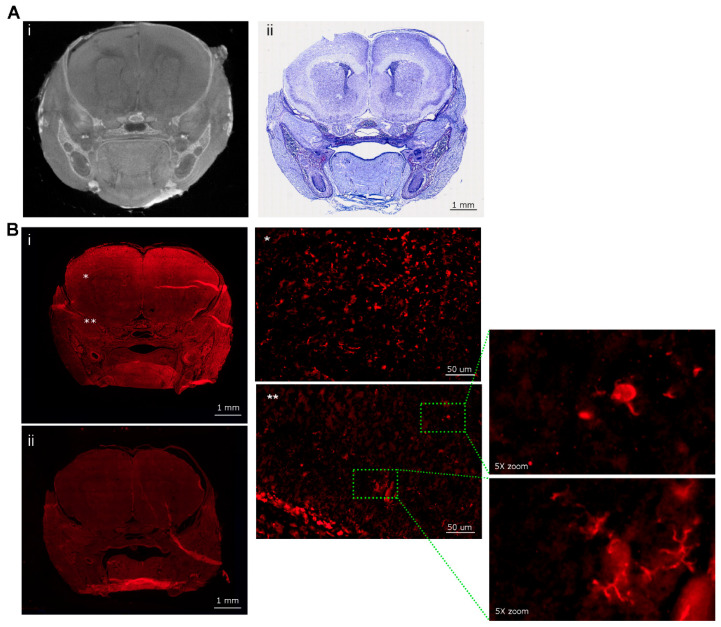
**Post-diceCT tissue analysis.** (**A**) (**i**) diceCT analysis; (**ii**) tissue was destained, embedded, and coronal sections from the midbrain analysed by Nissl staining; (**B**) (**i**) immunofluorescent analysis for Iba1; (**ii**) no antibody control; images were collected using a VS200 slide scanner. An optical zoom is shown at * in the hippocampus and ** the outer cortex from section (**i**), with a further zoom of regions as indicated by green dotted boxes and lines.

**Table 1 viruses-16-01330-t001:** **Summary of mice at 10 dpi.** Mice in a single litter were left uninfected or ZIKV-infected and blindly allocated mouse ID 1–6; movement was analysed, weighed, and heads measured at 10 dpi. ZIKV infection status was defined retrospectively by RT-PCR. * *p* < 0.05, Student’s unpaired *t*-test.

**Mouse ID**	**Weight (g)**	**Head Length, L, Width, W, Depth, D (cm)**	**ZIKV RT-PCR**	**Movement**
1	6.40	L = 1.9; W = 0.8; D = 0.6	positive	Mild hind limb dysfunction
2	7.50	L = 2.3; W = 1; D = 1	negative	normal
3	6.64	L = 2.1; W = 1.2; D = 0.9	positive	Mild hind limb dysfunction
4	5.93	L = 2.1; W = 0.9; D = 0.7	positive	Mild hind limb dysfunction
5	7.37	L = 1.8; W = 0.7; D = 0.8	negative	normal
6	7.34	L = 2.1; W = 0.9; D = 0.7	negative	normal
**Mean ± SEM**	**Body Weight (g)**	**Head Length (cm)**	**Head Width (cm)**	**Head Depth (cm)**
Mock (2, 5, 6)	7.40 ± 0.05	2.17 ± 0.07	1.03 ± 0.09	0.87 ± 0.09
ZIKV (1, 3, 4)	* 6.32 ± 0.21	1.93 ± 0.09	0.8 ± 0.06	0.7 ± 0.06
	*p = 0.007, t = 5.042, df = 4*	*p = 0.102, t = 2.211, df = 4*	*p = 0.091, t = 2.214, df = 4*	*p = 0.189, t = 1.581, df = 4*

**Table 2 viruses-16-01330-t002:** **Summary of blinded scoring of mixed 6 dpi cohort.** UI = uninfected; UD = undetermined. Unequivocal assignments for ZIKV are highlighted.

Mouse ID	Reviewer 1	Reviewer 2	Reviewer 3	Reviewer 4
1	No pathological findings were noted	No pathological findings were noted	No pathological findings were noted	No pathological findings were noted
	**UI**	**UI**	**UI**	**UI**
2	Major gap at the rear of the brain	Asymmetrical gaps at the level of the hippocampus/thalamus	Asymmetrical, apparent volume loss in the hippocampus	Asymmetrical, increased space between the hippocampal region and thalamus and towards the hindbrain
	**ZIKV**	**ZIKV**	**ZIKV**	**ZIKV**
3	No pathological findings were noted	Some gaps, but no asymmetrical volume loss	No pathological findings were noted	Slight gaps between the hippocampal region and thalamus
	**probably UI**	**UI**	**UI**	**probably UI**
4	Multiple asymmetrical gaps penetrating across the midbrain	Clear asymmetrical gaps between structures, with the majority of abnormalities around the thalamus and hippocampus, continuing through the posterior brain	Very clear asymmetrical abnormalities, particularly at the level of the posterior hippocampus, and continuing more posteriorly	Gaps between hippocampus and cortex (right) and thalamus (left); Gap (right) appears to go through the majority of the brain
	**ZIKV**	**ZIKV**	**ZIKV**	**ZIKV**
5	Maybe minor anomalies;	Asymmetrical gaps between the hippocampus and thalamus	Abnormality in the region anterior to nuc accumbens and at the level of striatum (less defined); Clear asymmetry at the level of the poster hippocampus;	Gap between the hippocampal region and the thalamus
	**UD**	**ZIKV**	**ZIKV**	**probably ZIKV**
6	Symmetrical gaps penetrating across images through the midbrain	Clear gaps at the hippocampus, continuing throughout the posterior brain	Very clear abnormalities, particularly at the level of the posterior hippocampus and extending posteriorly	Gap between the hippocampal region and thalamus on the left and right side, which runs through the majority of the brain
	**ZIKV**	**ZIKV**	**ZIKV**	**ZIKV**
7	Maybe minor anomalies	Minor abnormalities	Some minor abnormality, which may be an imaging artifact or normal anatomical variation	Slight gap between the hippocampal region and thalamus;
	**UD**	**UI**	**UD**	**probably UI**

**Table 3 viruses-16-01330-t003:** Quantitation of changes in volumes of different regions of the brain at 6 dpi in a blind cohort of uninfected (*n* = 3) and ZIKV-infected (*n* = 4) mice. Numbers 1–7 represent individual mice. See Figure 2 for brain region; values represent mm^3^. SD = standard deviation; % variation from the mean = (SD/mean) * 100; heat map scale is shown.

Mouse Number	Total Brain	1. Olfactory Bulb	2. Cerebrum	3. Hippocampus	4. Interbrain	5. Midbrain	6. Cerebellum	7. Hindbrain
**1**	14,940	974	6985	878	1922	1221	1437	1522
**2**	15,549	786	8429	549	1408	1486	1378	1513
**3**	15,534	787	8665	726	1377	1174	1385	1421
**4**	15,677	775	8604	892	1312	1210	1353	1531
**5**	15,565	774	7845	974	1794	1126	1406	1646
**6**	16,045	822	8412	861	1699	1347	1489	1416
**7**	15,402	718	8397	623	1569	1198	1490	1407
**SD**	305	75	550	145	214	114	50	80
**mean**	15,530	805	8191	786	1583	1252	1420	1494
**% variation from the mean**	1.9	9.3	6.7	18.4	13.5	9.1	3.5	5.3
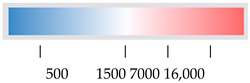								

## Data Availability

The data presented in this study are openly available in the Figshare repository at https://figshare.com/articles/dataset/diceCT_imaging_of_the_ZIKV_mouse_brain_during_development/26086651, edited 19 August 2024 and include video compilations and data files for 3D slicer segmentation analysis.

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
