# Peer review of "Application of diceCT to Study the Development of the Zika Virus-Infected Mouse Brain"

_viruses, 2024, doi:10.3390/v16081330_

Round 1

Reviewer 1 Report

Comments and Suggestions for Authors

Comments on the Quality of English Language

There are minor typographic mistakes that can easily be remedied.

Reviewer 2 Report

Comments and Suggestions for Authors

This is a generally well written manuscript describing application of an interesting and useful imaging technique to an important viral disease using neonatal mice.  I do not have substantive criticisms or suggestions other than to edit the first sentence of the introduction:

Capitalize Flavivirus

Change "where ..." to something like "responsible for clinically significaant outbreaks that culminated in .."

Comments on the Quality of English Language

English is fine 

Round 2

Reviewer 1 Report

Comments and Suggestions for Authors

My original comments and suggestions have been adequately addressed in this revision, except for one point. I suggested to convert the unit of volume from um3 to mm3. Indeed, I see that the authors changed the units. However, the conversion factor they used is incorrect. 1x108 uum3 = 0.1 mm3; so I don't understand why the values in Fig. 2C, Fig. 3, and Table 3 are way off... For context, 1,000 mm3 = 1 ml, so if you're showing brain volumes that are 104 mm3, it means the 10-day old pup has a brain volume of 10 ml, which is not possible. Please confirm that the values reported in the manuscript have been converted and reported correctly.

Comments on the Quality of English Language

looks OK

Author Response

Comment 1: 

My original comments and suggestions have been adequately addressed in this revision, except for one point. I suggested to convert the unit of volume from um3 to mm3. Indeed, I see that the authors changed the units. However, the conversion factor they used is incorrect. 1x108 uum3 = 0.1 mm3; so I don't understand why the values in Fig. 2C, Fig. 3, and Table 3 are way off... For context, 1,000 mm3 = 1 ml, so if you're showing brain volumes that are 104 mm3, it means the 10-day old pup has a brain volume of 10 ml, which is not possible. Please confirm that the values reported in the manuscript have been converted and reported correctly.

Rebuttal: We apologise, this was incorrectly calculated and has been amended and correct scales presented in Figure 2 and 3.